# Neonatal Bloodstream Infection with Ceftazidime-Avibactam-Resistant *bla*_KPC-2_-Producing *Klebsiella pneumoniae* Carrying *bla*_VEB-25_

**DOI:** 10.3390/antibiotics12081290

**Published:** 2023-08-05

**Authors:** Charalampos Zarras, Elias Iosifidis, Maria Simitsopoulou, Styliani Pappa, Angeliki Kontou, Emmanuel Roilides, Anna Papa

**Affiliations:** 1Microbiology Department, Hippokration Hospital, 54642 Thessaloniki, Greece; zarraschak6@gmail.com; 2Department of Microbiology, School of Medicine, Faculty of Health Sciences, Aristotle University of Thessaloniki, 54124 Thessaloniki, Greece; s_pappa@hotmail.com (S.P.); annap@auth.gr (A.P.); 3Infectious Disease Unit, 3rd Department of Pediatrics, School of Medicine, Faculty of Health Sciences, Hippokration Hospital, 54642 Thessaloniki, Greece; simitsop@auth.gr (M.S.); roilides@auth.gr (E.R.); 4Basic and Translational Research Unit, Special Unit for Biomedical Research and Education, School of Medicine, Faculty of Health Sciences, Aristotle University of Thessaloniki, 54124 Thessaloniki, Greece; 51st Department of Neonatology, School of Medicine, Faculty of Health Sciences, Aristotle University of Thessaloniki, 54124 Thessaloniki, Greece; angiekon2001@yahoo.gr

**Keywords:** multidrug resistance, Gram-negative bacteria, *Enterobacterales*, carbapenemases, *bla*_VEB-25_ carbapenemase, neonatal intensive care unit

## Abstract

Background: Although ceftazidime/avibactam (CAZ/AVI) has become an important option for treating adults and children, no data or recommendations exist for neonates. We report a neonatal sepsis case due to CAZ/AVI-resistant *bla*_KPC-2_-harboring *Klebsiella pneumoniae* carrying *bla*_VEB-25_ and the use of a customized active surveillance program in conjunction with enhanced infection control measures. Methods: The index case was an extremely premature neonate hospitalized for 110 days that had been previously treated with multiple antibiotics. Customized molecular surveillance was implemented at hospital level and enhanced infection control measures were taken for early recognition and prevention of outbreak. Detection and identification of *bla*_VEB-25_ was performed using next-generation sequencing. Results: This was the first case of a bloodstream infection caused by KPC-producing *K. pneumoniae* that was resistant to CAZ/AVI without the presence of a metalo-β-lactamase in the multiplex PCR platform in a neonate. All 36 additional patients tested (12 in the same NICU and 24 from other hospital departments) carried wild-type *bla*_VEB-1_ but they did not harbor *bla*_VEB-25_. Conclusion: The emergence of *bla*_VEB-25_ is signal for the horizontal transfer of plasmids at hospital facilities and it is of greatest concern for maintaining a sharp vigilance for the surveillance of novel resistance mechanisms. Molecular diagnostics can guide appropriate antimicrobial therapy and the early implementation of infection control measures against antimicrobial resistance.

## 1. Introduction

Antimicrobial resistance (AMR) is a public health threat facing humanity as it tests the resilience of health systems worldwide [1,2]. Various genetic elements are associated with the development of resistance because they manage via complex pathways to be transmitted between bacteria [3]. In addition, other practices such as delayed and/or incorrect diagnosis and the prescription of broad-spectrum antibiotics reinforce the problem of AMR [4]. Advances and innovations in the whole genome sequencing method and the bioinformatics revolution contribute to the immediate detection of the causes of resistance and the taking of timely and effective control measures [5].

A decisive factor in the development of AMR in healthcare facilities and especially in the intensive care units (ICU) of hospitals is the spread of multiresistant Gram-negative bacteria. *Enterobacterales* are the most important, among which *Klebsiella pneumoniae* is the main representative. *K. pneumoniae* is the second most common Gram-negative opportunistic pathogen and one of the most prevalent causes of community- and hospital-acquired infections [6]. It is responsible for health-care-associated pneumonia [7] and bacterial neonatal sepsis in low- and middle-income countries [8]. A serious public health threat is the emergence and dissemination of carbapenem-resistant *K. pneumoniae* (CRKP) that is associated with high morbidity and mortality, increased medical costs, and prolonged hospital stay [9]. Ιn addition, CRKP infections affect disability-adjusted life years (DALYs) per 100,000 population with a median value in the European Union of 11.5 years, while for these infections treatment options are limited [10,11]. CRKP isolates have a variety of mechanisms, which may confer resistance to virtually all available β-lactam antibacterial drugs, including carbapenems. The main resistance molecular mechanism is the production of a range of carbapenemases, including KPC, NDM, VIM, and OXA-48-like carbapenemases [12,13]. KPC-producing CRKP strains display the most extensive global distribution and represent a significant challenge due to their limited therapeutic options [14].

A novel β-lactam/β-lactamase inhibitor (BL/BLIs) combination is effective against strains of non-metallo-β lactamase producing *Enterobacterales* (Ambler class A, class C, and some class D β-lactamases) [15,16]. Ceftazidime/avibactam (CAZ/AVI) [17] has become an important first-line option for treating adult and pediatric (>3 months of age) patients with serious infections caused by carbapenem-resistant organisms, but not yet for neonates (IDSA) [18]. It is indicated for the treatment of complicated intra-abdominal and urinary tract infections, and infections caused by carbapenem-resistant *Enterobacterales* (CRE) or carbapenem-resistant *Pseudomonas aeruginosa*, in patients with limited or no other treatment options [19].

Although KPC-producing *Enterobacterales* strains are generally considered susceptible to CAZ/AVI, isolates resistant to this antimicrobial agent have been documented without the evidence of metallo-β-lactamases [20]. Ιn 2018, a rapid risk assessment conducted by ECDC identified CAZ/AVI resistance in CRE as a public health threat that merits careful monitoring [21]. CAZ/AVI resistance mechanisms include the increased expression of the *bla*_KPC_ gene product (acquisition of resistance was mostly associated with isolates harboring the substitution D179Y in *bla*_KPC-3_ or in *bla*_KPC-2_) [22,23], the presence of other genetic determinants of resistance against ESBL-producing *Enterobacterales* (SHV-, CTX-M-, or VEB-type β-lactamases) [24,25], changes in cell permeability (i.e., non-functional porins- OmpK35, OmpK36, and OmpK37) [26], and the expression of efflux pumps [27].

VEB-type β-lactamases (Vietnamese extended-spectrum β-lactamase) are a group of Ambler class A enzymes inhibited by avibactam. *bla*_VEB-25_ differs from *bla*_VEB-1_ by a missense mutation (substitution of lysine with arginine at position 237 -K234R) [28], which compromises the inhibitory efficiency of avibactam [29].

Herein, we report a successful treatment of bloodstream infection associated with CAZ/AVI-resistant *bla*_KPC-2_-producing *K. pneumoniae* carrying *bla*_VEB-25_ in a preterm neonate hospitalized in the neonatal intensive care unit (NICU) of a tertiary hospital and the use of a customized active surveillance program in conjunction with infection control measures for the early recognition and prevention of an outbreak.

## 2. Results

### 2.1. Index Case

The index case was the first neonate of a twin pregnancy born to a 33-year-old healthy primigravida at gestational age of 25w^+5d^ (birth weight = 850 gr, appropriate for a gestational age neonate) due to the premature rapture of membranes and the onset of labor. Postnatally, the patient presented respiratory distress syndrome, patent ductus arteriosus, severe bronchopulmonary dysplasia and need for prolonged mechanical ventilation, posthemorrhagic ventricular dilation, gastro-oesophageal reflux disease, retinopathy of prematurity, and episodes of late onset sepsis (LOS). The first LOS occurred on the fourth day of life due to carbapenem-resistant *Acinetobacter baumannii,* which was successfully treated. The patient was colonized with carbapenem-resistant *A. baumannii* and *Providencia stuartii* between Day 4 and 25, respectively. During that time, the neonate had been exposed to multiple antibiotic regimens for prolonged time periods, including meropenem, aminoglycosides, colistin, tigecycline, and CAZ/AVI due to episodes of suspected LOS and colonization by CR Gram-negative bacteria.

At Day 108, the neonate was on nasal continuous positive airway pressure due to chronic lung disease, and presented with fever and impaired peripheral perfusion. Empiric antibiotic treatment with colistin (300,000 IU/kg/day every 8 h), tigecycline (2 mg/kg/day every 12 h) and daptomycin (10 mg/kg/day once daily) was immediately initiated for suspected sepsis and due to the previous administration of multiple antimicrobial regimens. Blood culture was positive for a Gram-negative rod within 24 h since the onset of symptoms. A multiplex PCR platform (Biofire^®^ FilmArray^®^, Biomeriuex, Marcy-l’Étoile, France) was used within an hour from positive blood culture. A *bla*_KPC_ producing *K. pneumoniae* was detected and CAZ/AVI at a reduced dose of 31 mg/kg/d every 8 h was added to the antimicrobial regimen in attendance of the Antimicrobial Susceptibility Testing (AST).

During the first 48 h of this sepsis episode, the neonate deteriorated, requiring mechanical ventilation and possessing high inflammatory indices (max CRP value of 394 mg/L) and thrombocytopenia. At Day 110, the AST displayed a high level of resistance to almost all antimicrobial agents, including piperacillin/tazobactam, cefepime, cefoxitin, ceftazidime, ceftriaxone, imipenem, meropenem (MIC ≥ 16 mg/L), amikacin, gentamicin, ampicillin/sulbactam, aztreonam, ciprofloxacin, levofloxacin, fosfomycin, and trimethoprim/sulfamethoxazole. It was also resistant to novel agents, like ceftolozane/tazobactam and CAZ/AVI, while it was only susceptible to tigecycline and colistin. The isolate displayed a positive phenyl boronic acid phenotypic test and the lateral flow immunoassay, and the PCR method confirmed that the isolate carried *bla*_KPC_.

A favorable clinical and microbiological response was documented including defervescence and a decrease in CRP within 48–72 h, the first negative blood culture within 4 days, and the discontinuation of invasive mechanical ventilation within 8 days of colistin and tigecycline initiation. The administration of both daptomycin and CAZ/AVI was discontinued, whereas ciprofloxacin was empirically added four days after the first positive blood culture for a total of 13 days. The neonate was successfully treated with colistin and tigecycline for a total of 18 days.

#### NGS Report

A variety of genes conferring resistance to antimicrobial agents and heavy metals, as well as genes related to virulence, capsule, and efflux, and regulator systems were detected (Table 1). Only one serine-carbapenemase was detected, which was the *bla*_KPC-2_ gene and belonged to ST35. Another five β-lactamases (*bla*_SHV-33_, *bla*_TEM-1B_, *bla*_VEB-25_, *bla*_DHA-1_, and *bla*_OXA-10_) were co-detected, including the *bla*_VEB-25_. The co-production of *bla*_KPC-2_ and *bla*_VEB-25_ in *K. pneumoniae* has been associated with CAZ/AVI resistance in the absence of metallo-β-lactamase [24].

### 2.2. Molecular and Phenotypic Surveillance within the NICU and the Hospital

Thirteen *K. pneumoniae* strains were isolated from stool samples of neonates hospitalized in the NICU within a period of 3 months upon the recognition of the index case. Among these isolates, only the index case was *bla*_VEB-25_ positive (Figure 1A), confirming the NGS result. Based on the AST results, 24 additional carbapenem-resistant *K. pneumoniae* strains collected from various hospital sites were also analyzed with targeted PCR; even though they contained *bla*_VEB-1_, they did not harbor *bla*_VEB-25_ (Figure 1B).

Based on the AST results of the 24 carbapenem-resistant *K. pneumoniae* strains collected from various hospital sites, half were characterized as pan-drug-resistant [PDR, non-susceptibility to all agents in all antimicrobial categories (i.e., bacterial isolates are not susceptible to any clinically available drug)], and the other half as extensively drug resistant [XDR, non-susceptibility to at least one agent in all but two or fewer antimicrobial categories (i.e., bacterial isolates remain susceptible to only one or two antimicrobial categories)]. Therefore, all 24 CRKP isolates displayed high levels of resistance to almost all antimicrobials including imipenem (MIC ≥ 16 mg/L), meropenem (MIC ≥ 16 mg/L), amikacin (MIC ≥ 16 mg/L), gentamicin (MIC ≥ 16 mg/L), ampicillin/sulbactam (MIC ≥ 32 mg/L), piperacillin/tazobactam (MIC ≥ 128 mg/L), aztreonam (MIC ≥ 64 mg/L), cefepime (MIC ≥ 64 mg/L), cefoxitin (MIC ≥ 64 mg/L), ceftazidime (MIC ≥ 64 mg/L), ceftriaxone (MIC ≥ 64 mg/L), ciprofloxacin (MIC ≥ 4 mg/L), levofloxacin (MIC ≥ 8 mg/L), fosfomycin (MIC ≥ 256 mg/L), and trimethoprim/sulfamethoxazole (MIC ≥ 320 mg/L). These isolates were also analyzed with targeted PCR; even though they contained *bla*_VEB-1_, they did not harbor *bla*_VEB-25_.

### 2.3. Overall Assessment

This index case was the last neonate that was infected with *A. baumannii* and colonized by *P. stuartii* within the NICU after the implementation of enhanced infection control measures targeting these two pathogens. Upon the recognition of the first *K. pneumoniae* producing *bla*_KPC-2_ and *bla*_VEB-25_ and a combination of intensified and targeted infection control actions in the unit, there were no other cases within the NICU for the next 6 months.

## 3. Discussion

We report a neonatal case of a bloodstream infection caused by a *K. pneumoniae* strain co-producing *bla*_KPC-2_ and *bla*_VEB-25_ β-lactamases and emphasize the use of precise medicine to customize infection control measures. Treatment options for infections caused by carbapenem-resistant bacteria are extremely limited in neonates. The “off label” use of either “last-line” antimicrobial agents (such as polymyxins and tigecycline) or the currently available newer β-lactam/β-lactam inhibitor combinations, such as CAZ/AVI, meropenem-vaborbactam, and imipenem-cilastatin-relebactam that are not yet licensed for neonates, for the empirical treatment of neonatal sepsis in areas endemic for CRKP is still questionable due to limited pharmacokinetic data and local epidemiology of resistant genes [30].

One of the mechanisms that confers resistance to CAZ/AVI is the new *bla*_KPC_ variants that are constantly appearing worldwide. Very recently, Shi et al. reported multiple novel variants in a *K. pneumoniae* strain carrying *bla*_KPC-2_ from two separate patients during their exposure to CAZ/AVI. In one patient, the *bla*_KPC-2_ mutated to *bla*_KPC-35_, *bla*_KPC-78_, and *bla*_KPC-33_ during the same period, while in the other patient it mutated to *bla*_KPC-79_ and *bla*_KPC-76_, thus enhancing the level of resistance [31]. ST258 *K. pneumoniae* is considered the most frequent type in the majority of *bla*_KPC_-associated infections resistant to CAZ/AVI [32].

The *bla*_KPC-2_-harboring *K. pneumoniae* isolated in our study belonged to Sequence Type ST35. To the best of our knowledge, this is the first report of ST35 CRKP bearing both *bla*_KPC-2_ and *bla*_VEB-25_ that confers resistance to CAZ/AVI. Findlay et al. identified two isolates as belonging to Sequence Types ST147 and ST258, harboring *bla*_VEB-25_ on the plasmid, that confer resistance to CAZ/AVI [33].

To date, there are three reports of CAZ/AVI-resistant KPC-producing *K. pneumoniae* emergence in Greece, all in adults (six infected and five colonized patients) [24,34,35]. Notably, the first CAZ/AVI-resistant clinical isolate was detected in Greece before the introduction of CAZ/AVI in clinical practice. The resistance was due to the existence of *bla*_KPC-23_ (variant differed from *bla*_KPC-3_ by one -V240A, and from *bla*_KPC-2_ by two amino acid substitutions -V240A and H274Y) [34]. CAZ/AVI resistance due to the harboring of *bla*_VEB-25_ has been reported in two additional cases (one isolate from blood and one from the lower respiratory tract) from patients without prior CAZ/AVI exposure [35]. Eight more CAZ/AVI-resistant CRKP isolates were detected in patients not previously exposed to CAZ/AVI (two patients with catheter-related bloodstream infections, one with ventilator-associated pneumonia, and five with colonization); the resistance was conferred by the harboring *bla*_VEB-25_ and *bla*_VEB-14_ [24]. After intense epidemiological and microbiological surveillance in our NICU, as well as in pediatric and adult departments within our general hospital (especially pediatric and adult intensive care units), we could not find the source of this resistant organism. However, our index patient had been previously exposed to multiple courses of antimicrobial agents, including CAZ/AVI, and also had gut colonization with XDR Gram-negative bacteria, such as *A. baumannii* and *P. stuartii*.

This was the first premature neonate presenting with sepsis due to CAZ/AVI-resistant *bla*_KPC-2_-harboring *K. pneumoniae* carrying the *bla*_VEB-25_ that was successfully treated with non-conventional “off-label” antimicrobial agents. Currently, available diagnostic platforms detect the presence of the most prevalent carbapenemases, such as KPC, VIM, NDM, and OXA. Neonatologists and infectious disease specialists should be cautious when interpreting the results from these molecular platforms for decision making in empiric and targeted treatment for neonatal sepsis. The mechanism of resistance, especially for the newer β-lactam/β-lactamase inhibitors, may differ in times and in different parts of the world and even within the same institution [36]. In addition, various mechanisms of CAZ/AVI resistance emphasize the need for the surveillance of CAZ/AVI-resistant pathogens, as well as for its judicious use.

## 4. Materials and Methods

### 4.1. Risk Assessment and Bundle of Actions Taken after Index Case

This was the first case of a bloodstream infection caused by KPC-producing *K. pneumoniae* that was resistant to CAZ/AVI without the presence of metalo-β-lactamase in the multiplex PCR platform in a neonate. The bundle of actions implemented is summarized in Figure 2 and included: (1) enhanced infection control measures including strict isolation of the case index; (2) continuation of active surveillance for CRE and tests for CAZ/AVI susceptibility reported for all isolates recovered from surveillance; (3) application of next-generation sequencing (NGS) and molecular testing for the index case to identify probable mechanism(s) of CAZ/AVI resistance; (4) targeted PCR analysis in all CRE isolates from all neonates in the ICU, independently to CAZ/AVI susceptibility; and (5) targeted PCR analysis specifically for CAZ/AVI-R isolates from other departments of the hospital to identify potential sources and/or burden of a potential outbreak.

#### 4.1.1. Infection Control Measures

The NICU was already on strict infection control measures, including the cohorting of all neonates colonized/infected with an XDR *A. baumannii* strain. Upon recognition of this index case, extra measures were taken: isolation of index case, dedicated nurse for all shifts, universal application of contact precautions, written reports of active surveillance, and daily audits by infection control team (with a dedicated infection control nurse and a dedicated pediatric infectious disease specialist).

#### 4.1.2. Active Surveillance

Already in place with twice weekly colonization cultures. Specifically, stool samples were taken from the neonates on the NICU and cultured on MacConkey agar plates supplemented with 1 mg/L meropenem. AST was applied to all isolates as written in Section 4.2, including CAZ/AVI susceptibility. Active surveillance included not only gut and pharyngeal colonization but also environmental cultures.

### 4.2. Microbiological Methods, Antimicrobial Susceptibility Testing, and Phenotypic Analysis

CRKP was identified with a VITEK 2 automated system (Biomeriuex, Marcy-l’Étoile, France) using the GN ID according to the manufacturer’s instructions. The AST of *K. pneumoniae* was performed using the AST 376 and XN10 cards; the interpretation of results was according to the European Committee on Antimicrobial Susceptibility Testing (EUCAST) breakpoints of January 2022. Susceptibility testing to CAZ/AVI was performed using MIC test strips (Liofilchem srl, Roseto, Italy), while susceptibility testing to colistin was performed using the broth microdilution method (Liofilchem srl, Roseto degli Abruzzi, Italy). Tigecycline was evaluated using the susceptibility breakpoints approved by the US Food and Drug Administration (MIC ≤ 2 mg/L for susceptibility and ≥8 mg/L for resistance).

The isolate was phenotypically tested for KPC and metallo-β-lactamase (MBL) production using phenylboronic acid and ethylenediaminetetraacetic acid. Carbapenemase genes *bla*_KPC_, *bla*_NDM_, *bla*_OXA-48-like_, *bla*_IMP_, and *bla*_VIM_ were screened with a multiplex lateral flow immunoassay (NG-Test CARBA 5, NG Biotech, Guipry, France). The detection limits using purified recombinant enzymes for NDM, KPC, IMP, VIM, and OXA-48-like were 150, 600, 200, 300, and 300 pg/mL, respectively.

### 4.3. Next-Generation Sequencing (NGS)

DNA was extracted using the DNA extraction kit (Qiagen, Hilden, Germany). The Qubit double-strand DNA HS assay kit (Q32851, Life Technologies Corporation, Grand Island, NY, USA) was used for measuring the dsDNA concentration. All procedures regarding shearing, purification, ligation, barcoding, size selection, library amplification and quantitation, emulsion PCR, and enrichment were conducted according to the manufacturer’s guidelines. After template enrichment, sequencing was performed on an Ion PGM™ semiconductor sequencer using a Hi-Q View Sequencing Kit and a 316 Chip V2 BC (Thermo Fisher Scientific, Waltham, MA, USA). The sequence reads were de novo assembled and annotated using Geneious Prime version 2021.2.1. The sequence of the *K. pneumoniae* NTUH-K2044 strain (Accession number NC-012731) was used as reference.

### 4.4. MLST and Detection of Antimicrobial Resistance Genes and Plasmids

MLST and antimicrobial resistance genes and plasmids were identified using the online databases at the Center for Genomic Epidemiology (MLST-2.0, Resfinder 4.1 and Plasmid finder) [37,38,39,40,41,42,43,44] and the Comprehensive Antibiotic Resistance Database (CARD) Bait Capture Platform 1.0.0 [https://card.mcmaster.ca/ (accessed on 4 August 2023)]. Genes related to virulence, resistance to heavy metals, efflux, regulator systems, and capsules were detected using the Institut Pasteur website on *K. pneumoniae* [https://bigsdb.pasteur.fr/klebsiella/ (accessed on 4 August 2023)].

### 4.5. Targeted PCR Analysis

Molecular surveillance at the NICU and hospital level: After the recognition of the existence of *bla*_VEB-25_ as the mechanism of CAZ/AVI resistance in KPC *K. pneumoniae*, targeted PCR protocol was initiated to investigate transmission within the NICU, but also to other carbapenem-resistant *K. pneumoniae* isolated from other pediatric and adult departments in the hospital (particularly, pediatric and adult intensive care units). A total of 37 *K. pneumoniae* strains were tested for the presence of *bla*_VEB-1_. Thirteen of them were isolated from stool samples collected from neonates in the NICU where the *bla*_VEB-25_ index case was identified, and twenty-four strains were isolated from different clinical sources (blood, urine, tracheal aspirate, trauma, and central venous catheter) collected from several departments of the hospital to investigate potential sites of outbreak. Plasmid DNA was extracted using the alkaline lysis method, as described previously (H.C.Birnboim and J.Doly NAR 7: 1513-1523, 1979). For PCR amplification, VEB-F (5′-CGA CTT CCA TTT CCC GAT GC-3′) and VEB-B (5′-GGA CTC TGC AAC AAA TAC GC-3′) primers were used as diagnostic primers to amplify a 642 bp internal VEB-1 DNA segment, whereas the external primers VEBcas-F (5′-GTT AGC GGT AAT TTA ACC AGA TAG-3′) and VEBcas-B (5′-CGG TTT GGG CTA TGG GCA G-3′) were used to amplify the entire gene for DNA sequencing. For each PCR reaction, 50–70 ng of *K. pneumoniae* plasmid DNA was used in a standard PCR reaction using Kapa Hi Fi DNA polymerase (KAPA Biosystems) with the following amplification program: 1 cycle of 95 °C 3 min, 35 cycles of 20 s at 94 °C, 30 s at 55 °C, 30 s at 72 °C, and a final extension step of 1 min at 72 °C. The PCR products were Sanger sequenced. Nucleotide sequence analysis and pairwise alignments were performed using the National Center of Biotechnology Information website [https://www.ncbi.nlm.nih.gov accessed on 4 August 2023)].

## 5. Conclusions

Applying next-generation sequencing technology is crucial for guiding the prediction of underlying resistance mechanisms facilitating the study of the evolution and molecular epidemiology of multidrug-resistant pathogens, especially in endemic areas. The emergence of *bla*_VEB-25_ is a warning for the horizontal transfer of plasmids at hospital facilities, and it is of greatest concern for maintaining a sharp vigilance for the surveillance of novel resistance mechanisms. The use of molecular diagnostics may guide appropriate antimicrobial therapy and the early implementation of strict infection control measures, and therefore could play an important role in the fight against antimicrobial resistance.

## Figures and Tables

**Figure 1 antibiotics-12-01290-f001:**
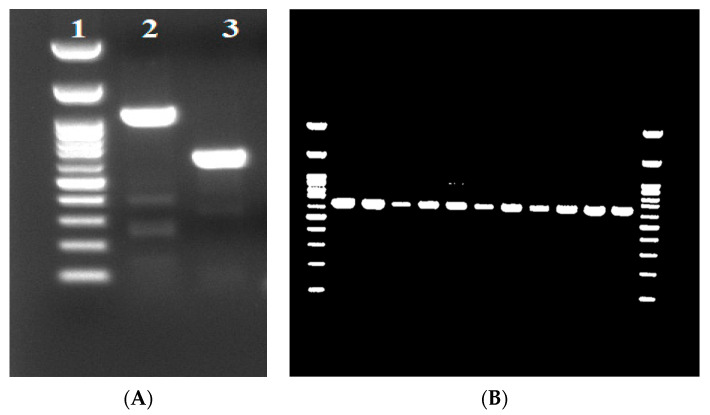
Agarose gel electrophoresis profile of the *bla*_VEB-25_ variant. Panel (**A**) shows the *bla*_VEB-25_ positive variant, whereas panel (**B**) shows the 642 bp amplified products of *bla*_VEB-25_-negative carbapenem-resistant *K. pneumoniae* strains. The amplified products of 1070 bp and 642 bp were produced using the external VEBcas-F/VEBcas-B (lane 2) and internal VEB-F/VEB-B primer pairs (lane 3). The amplified product containing the entire gene (1070 bp) was used to deduce the nucleotide sequence. The 100 bp DNA ladder with reference bands ranging from 100 bp to 1500 bp is indicated in lane 1.

**Figure 2 antibiotics-12-01290-f002:**
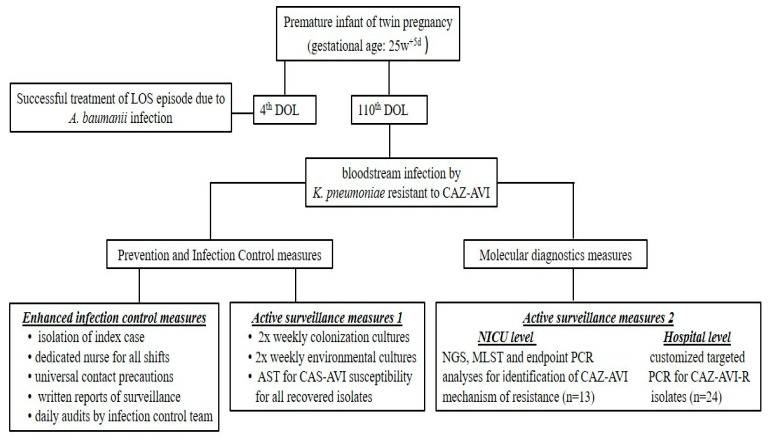
Summary of a bundle of actions followed in a premature neonate with a ceftazidime-avibactam-resistant KPC-2-producing *Klebsiella pneumoniae* bloodstream infection carrying the VEB-25 gene. LOS: late onset sepsis, DOL: day of life, CAZ-AVI: ceftazidime-avibactam.

**Table 1 antibiotics-12-01290-t001:** Genetic characteristics of the neonatal blood *K. pneumoniae* isolate of the study via NGS.

Strain ID		A1746/22
Date of isolation		25 February 2022
Biological sample		Blood
MLST		35
Plasmids		IncC, IncR, IncFIA(HI1), IncFIB(K), IncFIB(pKPHS1), IncFIB(pQil), IncFII(K)
Antibiotic Resistance	β-lactamases	SHV-33, TEM-1B, VEB-25, DHA-1, OXA-10
	Carbapenemases	KPC-2
	Aminoglycosides	ant(2″)-Ia, aph(3″)-Ib, aph(6)-Id, rmtB, aadA1
	Quinolone	qnrB4, oqxA, oqxB
	Fosfomycin	fosA
	Sulfonamide	sul1, sul2
	Phenicol	catA1, cmlA1
	Tetracycline	tet(A), tet(G)
Resistance to Heavy Metals		merC, merP, merT, silR
Virulence		kfuA, mrkA, mrkF, mrkH, mrkl, ybtE, ybtQ, ybtT, ybtX
Capsule		wzi
Efflux and Regulator Systems		acrR, envR, fis, marA, marR, oqxR, rob, sdiA, soxR, soxS, ramA, ramR, rarA

## Data Availability

The datasets generated during, and/or analyzed during, the current study are not publicly available due to the fact that these are the results of patient examinations carried out in a public hospital, but are available from the corresponding author on reasonable request.

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
