# Peer review of "Neonatal Bloodstream Infection with Ceftazidime-Avibactam-Resistant blaKPC-2-Producing Klebsiella pneumoniae Carrying blaVEB-25"

_antibiotics, 2023, doi:10.3390/antibiotics12081290_

Round 1

Reviewer 1 Report

The authors show an example of a troublesome variant bla gene VEB-25, and have done a whole genome sequence determination of the carrier strain, a Klebsiella pneumoniae. They also searched for the same gene in other carbapenem resistant isolates without finding any other example of VEB-25, only the related VEB-1, which is of course also a worrying finding of that resistance gene. The carrier strain was shown to carry multiple plasmids and a number of virulence determinants. 

While the results are interesting and worth publication, I find the presentation of results difficult to follow and interpret. The organisation of the paper with presenting results before the Materials and Methods section means the reader has to go back and forth to get all the important information. Relatively large parts of the text in Materials and Methods are essentially results, and what I especially found difficult was the description of the index case. Since this is important for the understanding of the paper I would suggest starting the Results section with a complete description of the findings from the index case. Now the information of drug treatment, and additional bacterial isolates is too scattered and most of the text in section 4.1 is better placed as Results. The information on isolation of other bacteria and consecutive drug treatments could be described in more detail.

Figure 1 is good, but since the same method apparently was used to detect VEB-1 in other isolates at least one (or more) examples of results from VEB-1 carrying strains would be good to have in the figure. 

Finally, the authors are worried about transmissibility of the VEB-25 gene. Here, I would have expected an attempt to link the particular gene with one of the plasmids found in the isolate, and that conjugation experiments would have been performed. Without these addition and considering the fact that no other isolate carrying VEB-25 was found, there is no real evidence for the transmissibility of the gene.

No real problems with understanding, however some improvements could be achieved by a professional language check

Reviewer 2 Report

The presented case is interesting as it raises the issue of antibiotic resistance in enterobacteria, Klebsiella pneumoniae in this case. The case is well-written and provides sufficient details regarding diagnostic and therapeutic management.

I have a few comments:

The introduction should begin with a section discussing the problem of antibiotic resistance in bacteria in general.

The resolution of Figure 2 needs improvement.

Two recent articles have been published that discuss the topic addressed by the authors and provide interesting information that I consider should be discussed in the Discussion section. (DOI: 10.1007/s10096-023-04582-0, DOI: https://doi.org/10.1128/spectrum.01714-21).

Minor editing of English language required. Please revise the manuscript once more.

Reviewer 3 Report

The manuscript titled "Neonatal bloodstream infection with ceftazidime-avibactam-resistant KPC-2 producing Klebsiella pneumoniae carrying the VEB-25 gene" is well-written and has the potential for publication with some revisions.

I noticed that the authors inconsistently use the term "bla" with VEB, but not with KPC. It would be beneficial to apply a consistent style for writing the names of these genes and italicize them for clarity.

Furthermore, the authors stated that they tested 37 K. pneumoniae strains for the presence of blaVEB-1. However, it would be helpful if they could provide an explanation as to why only K. pneumoniae strains were tested, considering that the gene can potentially be transmitted to other pathogenic microorganisms like E. coli through horizontal gene transfer. This clarification would enhance the understanding of the study's scope.

Additionally, the authors did not discuss the minimum inhibitory concentration (MIC) of various antimicrobial agents, particularly CAZ/AVI and carbapenems, among the bacteria producing blaVEB and blaKPC. Including this information would be valuable for assessing the resistance profiles of the bacteria.

The discussion section lacks comprehensiveness, as several significant findings have not been addressed. It would be beneficial to expand upon these findings and provide a more thorough analysis. Furthermore, the authors did not discuss the multilocus sequence types (MLST) in the manuscript and including this information would contribute to a more comprehensive understanding of the bacterial strains involved.

The English in the manuscript is generally acceptable, but there are a few grammatical and syntactical errors that need to be addressed.

Round 2

Reviewer 1 Report

This version is substantially improved. You may go through the language issue once more. One thing that escaped my notice last time is the misspelling of the first word in the abstract (Buckground should be Background). There may be other remaining minor issues. Much easier to read now.

Improved but small issues remain

Reviewer 3 Report

The manuscript is improved now and can be accepted